# Synthesis of ZnO Nanorods and Its Application in Zinc-Silver Secondary Batteries

Van Tu Nguyen [1,*], Hung Tran Nguyen [1] and Nu Huong Tran [2]

1   Institute of Chemistry and Materials, 17 Hoang Sam Street, Nghia Do, Cau Giay, Hanoi 100000, Vietnam
2   Faculty of Chemistry, Thai Nguyen University of Education, 20 Luong Ngoc Quyen Street, Thai Nguyen 250000, Vietnam
*   Correspondence: nguyenvantu882008@gmail.com

**Abstract:** In this paper, ZnO nanorods were synthesized by the hydrothermal method and used as anodes for zinc-silver batteries. The Tafel and EIS curve analysis results show that ZnO nanorods have better anti-corrosion and charge transport properties than ZnO powders. At 0.1 C discharge conditions, the ZnO electrode exhibits more stable cycle efficiency than the powder electrode; after 25 cycles, the capacity is higher by 95%. The superior electrochemical performance is due to the ZnO nanorods having the ability to conduct electrons and increase the surface area. Therefore, the possible growth mechanism of ZnO nanorods has been investigated.

**Keywords:** ZnO; zinc-silver batteries; zinc electrode; hydrothermal; electrochemical performance

## 1. Introduction

Zinc oxide (ZnO) has a direct wide bandwidth and large exciton binding energy. Therefore, today, it is a material with many applications in the fields of fluorescence, photo-catalysis, pyroelectricity, gas sensing, electrochemistry and solar cells [1–3]. ZnO can be prepared by various methods: sol-gel, sputtering, gel combustion, co-precipitation and hydrothermal [4–8]. The morphologies of ZnO can vary from nanosphere, nanorod [4], multi-dimensional zigzag [5] to flower [6], depending on the synthesis process. Therefore, according to the application, we can choose the suitable synthesis method. ZnO nano-materials can be used in various applications in environmental treatment, sterilization, electronics and energy [9–16]. For energy and battery, it is used as an anode material for silver-zinc, zinc-nickel, zinc-air and lithium-ion batteries [17,18].

Nowadays, nano-sized materials have been widely studied and applied for batteries and fuel cells [19–23]. The results show that electrochemical processes' efficiency depends on the active material and surface area. If the decreasing particle size of active material increases surface area, increasing porosity, thus increasing the kinetic reaction, dispersion, and discharge/charge process [24]. ZnO is widely used as an anode in Zn-Ni and Zn-Ag batteries for energy and battery. However, it has extended its application to supercapacitors, Li-ion batteries, and zinc-air batteries [1,25–28].

Zinc-silver batteries are superior in energy density and safety, so they are still being researched and developed, even though Li-ion and metal hydride batteries are very developed. These batteries have a long, successful history; they have been used in military and aerospace applications due to their unique properties such as stability, high current and high safety [1,2,25–28]. Zinc-silver batteries have high specific energy (up to 300 Wh/kg) and volumetric energy density (up to 750 Wh/dm), low self-discharge rate (~5% per month) and stable voltage during the discharge. Figure 1 shows the principle and construction of a zinc-silver battery.

The reaction process of the zinc-silver system is the reduction of silver oxide and the oxidation of zinc. Silver oxide can exist in two forms AgO and $Ag_2O$. The chemical reactions occurring at the cathode for divalent AgO and monovalent $Ag_2O$ are:

$$2AgO + H_2O + 2e \rightleftarrows Ag_2O + 2OH^-, E^0 = +0.607 \text{ V}$$

$$Ag_2O + H_2O + 2e \rightleftarrows 2Ag + 2OH^-, E^0 = +0.345 \text{ V}$$

For the anode, the zinc electrode is oxidized by a dissolution reaction and precipitation occurs:

$$Zn + 4OH^- \rightleftarrows Zn(OH)_4{}^{2-} + 2e^-, E^0 = -1.249 \text{ V}$$

$$Zn(OH)_4{}^{2-} = ZnO + H_2O + 2OH^-$$

Overall reaction:

$$Ag_2O + Zn \rightleftarrows 2Ag + ZnO \text{ or } AgO + Zn \rightleftarrows Ag + ZnO$$

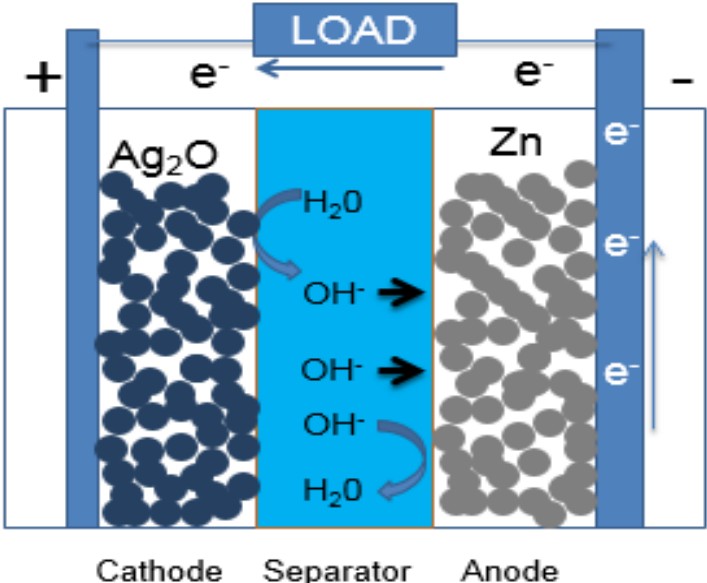

**Figure 1.** Zinc-silver battery with porous electrodes during discharge.

The limitations of zinc electrodes are poor rechargeability, where performance-limiting phenomena are attributed to dendrite growth, shape change, passivation, and hydrogen evolution (as in Figure 2). Therefore, the choice of solutions to increase the number of discharge/charge cycles for the battery on the basis of a zinc electrode is necessary. Moreover, nanotechnology is one of the most effective solutions. Recently, the use of nanoscale materials has been successfully demonstrated in rechargeable batteries.

Today, to combat the dendrite phenomenon of zinc, many solutions have been studied, such as using new electrodes, binders (electrodes), additives (electrolytes), and charging techniques [28–31]. The additives have been studied and introduced as oxides such as HgO, $Sb_2O_3$, $TiO_2$, $Pb_3O_4$, neodymium/lantana conversion coatings and $Y(OH)_3$ [32–36]. In addition, the electrolyte may add surfactants, gelating agents, phosphates and sodium tetraborate [37]. However, enhancing the battery properties is a challenging and ongoing process for the research community.

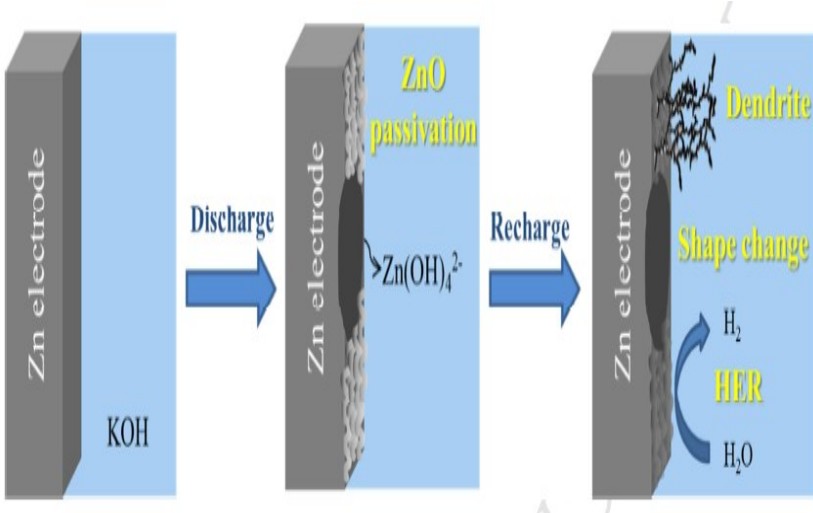

**Figure 2.** Schematic of the Zn electrode's dendrite phenomenon and the rechargeable limitation of the zinc electrode.

The ZnO nanorods have been used as anode materials for zinc and lithium-ion batteries. However, the drawbacks of zinc batteries are poor cycle life because of zinc electrode conformational change and cell short-circuits that occur when dendritic zinc needles grow during charge/discharge on the anode. Normally, ZnO powder is dissolved in an alkaline electrolyte and causes the dendritic phenomenon. ZnO nanorods limit this disadvantage, thus limiting the dendrite due to their high surface area, high density, and cycle stability [38–40]. In addition, it has also been studied and shown that ZnO nanorods have higher strength and stability than 1D structures. Therefore, recently, they have been widely studied and applied in zinc-silver, zinc-nickel, zinc-air, and lithium-ion batteries [41–47].

In the present work, we report the efficient solvent synthesis of ZnO nanorods and investigate the electrochemical and structural properties. The electrochemical performance of the as-synthesized samples has been studied, and improvement in electrochemical performance is explained based on Nyquist plots. The results indicate a significant improvement in zinc-silver batteries' shape change and life cycle.

## 2. Materials and Methods

### 2.1. Preparation Materials

ZnO nanorods were prepared by hydrothermal method from a mixture of $Zn(NO_3)_2$ 0.1M and NaOH 0.1M in $H_2O/C_2H_5OH$ (ratio $C_2H_5OH$: $H_2O$ = 1:1) at pH = 11. We have detailed the general conditions according to the document [7].

### 2.2. Characterization

The crystallographic structure of the sample was confirmed by powder X-ray diffraction spectroscopy (XRD, X'Pert Pro, Almelo, The Netherlands). The morphology of the samples was observed by the scanning electron microscope (SEM, S4800, JEOL, Tokyo, Japan). The specific surface area was determined by the Brunauer, Emmett and Teller (BET) method used on the Tri-Start 3000 instrument, Micromeritics. The sample size distribution was determined by laser scattering on the HORIBA Laser Scattered Particle Size Analyzer LA-950.

### 2.3. Assessment of Prepared ZnO Nanoparticles for Electrochemical Testing

The electrochemical properties of the materials ZnO nanorods were determined by electrochemical methods such as cyclic voltammetry, charge/discharge test and Electrochemical impedance spectroscopy (EIS) measurements. The anode electrode was prepared by the active materials ZnO nanorods (78 wt%), zinc powder (15 wt%), polyvinyl alcohol

binder (5 wt%), and mercury oxide (2 wt%) were mixed and pressed on silver wire current collecting grid at 15 MPa pressure. Electrolyte solution is 6M KOH saturated with ZnO. The anode electrode was dried in a vacuum oven at 80 °C for 10 h. The cathode electrode is prepared using sintered silver active materials. The cathode electrode has 300% capacity of the anode electrode. The electrodes were washed with deionized water until neutral. The silver electrode was air-dried, while the zinc electrode was dried in a vacuum oven at 80 °C for 10 h. The battery of nominal capacity is 2.0 Ah. The electrodes are assembled with a submerged electrolyte.

The following formula expresses theoretical battery capacity:

$$C = mnF/M$$

where: C is the battery capacity (Ah), m is the mass of the substance (g), n is the number of electrons per mole unit of the active substance (mol), F is the Faraday constant (96,485.34 $C.mol^{-1}$) and M is the molecular mass of the active substance ($g.mol^{-1}$). ZnO has a molecular weight of 81.36 $g.mol^{-1}$, which calculates the theoretical density of ZnO to be about 659 $mAh.g^{-1}$. Therefore, zinc electrode preparation was calculated, and the average mass of the active materials per electrode at about 30 g.

The electrochemical measurements were investigated at different potentials by Auto-lab Potentiostat (PGATAT30, Metrohm Autolab, Ultrecht, The Netherlands) system, and galvanostatic charge/discharge was by BTS-5 V/100 mA and 5 V/3 A battery-testing instrument (Neware, Hong Kong, China). First, all cells were discharged/charged 5 times at a constant current of 0.1 C for 20 h and discharged at a constant current of 0.1 C to a cut-off voltage of 1.40 V at room temperature. Then, cyclic voltammetry, Tafel plot and the electrochemical impedance spectroscope (EIS) tests were conducted in a conventional three-electrode system (Hg/HgO as the reference electrode, a commercial sintered Ag electrode as counter electrode, zinc electrode as working electrode). During the cyclic voltammetry test, the scan rate is 10 $mVs^{-1}$ and the sweep range is −0.85 to −1.65 V. Tafel curve, the sweep interval is related to the open circuit voltage ±100 mV with a rate sweep 0.5 mV $s^{-1}$. All the above electrochemical tests were performed at room temperature (25 ± 1 °C). And for comparison, the common ZnO anode material (purchased from Xiaoshan Chemical Factory, Hangzhou, China) is also fabricated by the same process.

## 3. Results and Discussion

### 3.1. Factors Affecting Morphology and Surface Structure of ZnO Nanoparticles

3.1.1. SEM Image Analysis

The results of the SEM image analysis are shown in Figure 3. The results figure shows that the zinc oxide sample has a nanostructure, length of 100 to 200 nm, and diameter of about 50 nm.

3.1.2. Determination of Surface Area by the BET Method

ZnO samples were measured by the BET method; $N_2$ isothermal adsorption at −196 °C. The $N_2$ adsorption and desorption isotherms are shown in Figure 4. The BET plot of the ZnO samples obtained from the software is also shown in Figure 5. The results show that the surface area of the ZnO samples is 17.05 $m^2/g$.

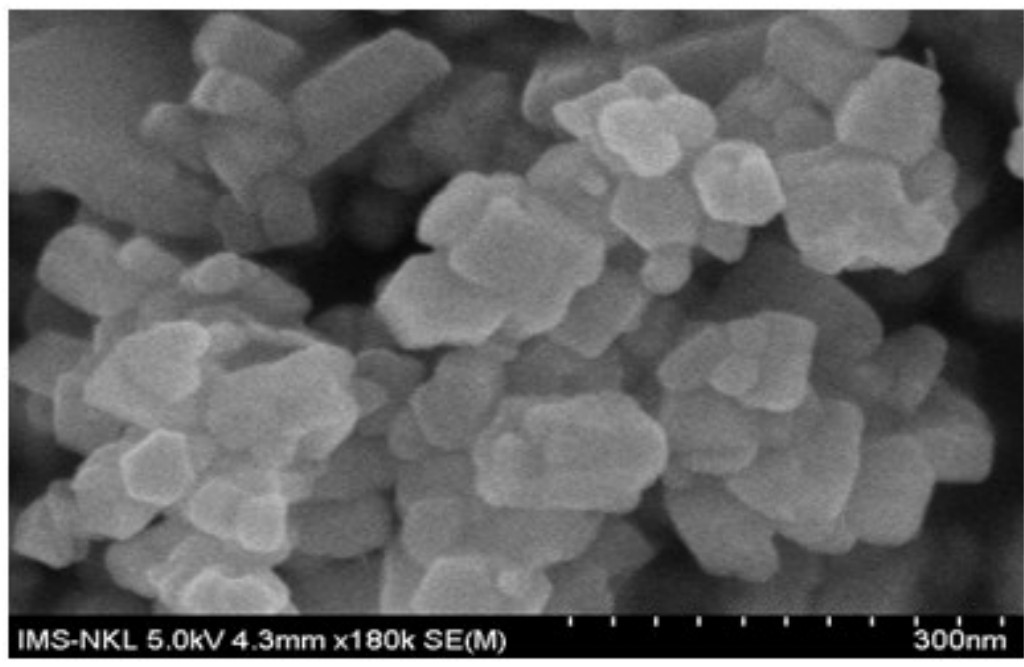

**Figure 3.** SEM image of ZnO nanomaterials.

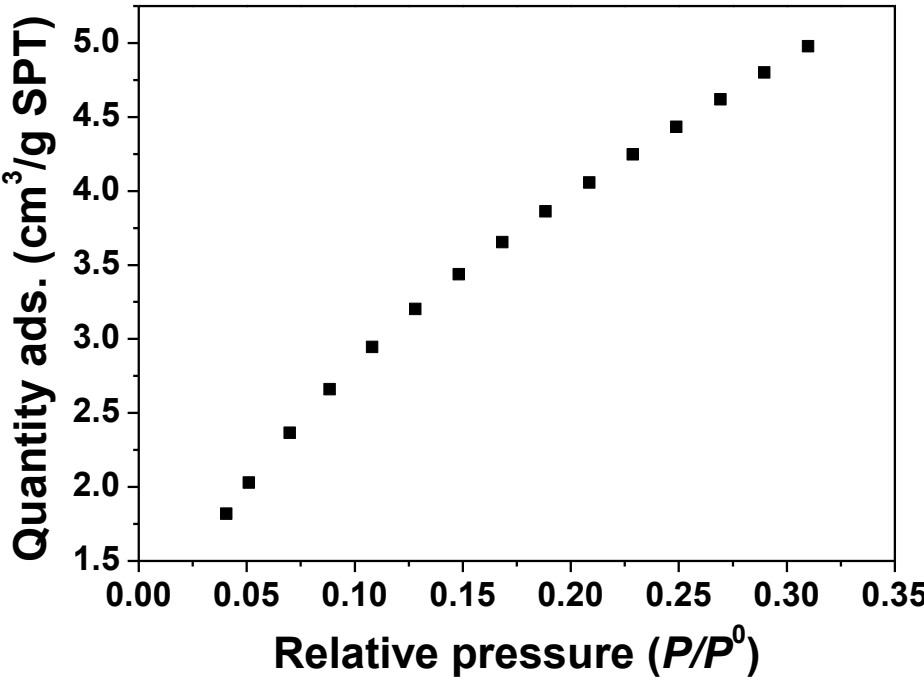

**Figure 4.** Adsorption isotherm. $N_2$ adsorption of ZnO samples, as-synthesis condition at a temperature of 180 °C.

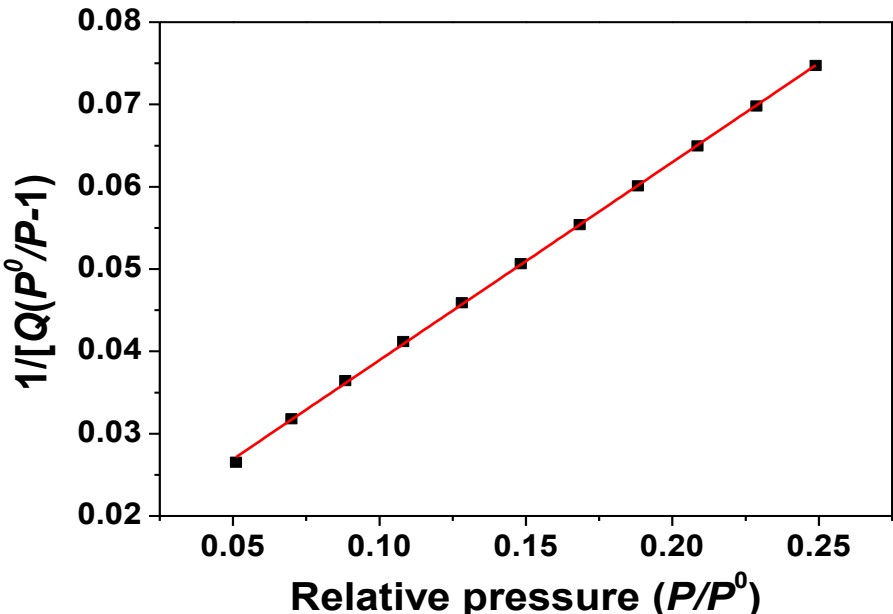

**Figure 5.** BET plot of as-synthesized ZnO nanorods obtained at a synthesis temperature of 180 °C.

### 3.2. Structure and Morphology

### 3.2.1. XRD Analysis

The XRD patterns of the ZnO samples are shown in Figure 6, showing the characteristic peaks at (100); (002); (101); (102); (110); (103); (112) and (201). This indicates the wurtzite structure of ZnO (plane of the ZnO phase, JCPDS file 36-1451). From the results shown, the diffraction peaks are narrow, sharp and symmetrical, indicating that the samples are well crystalline.

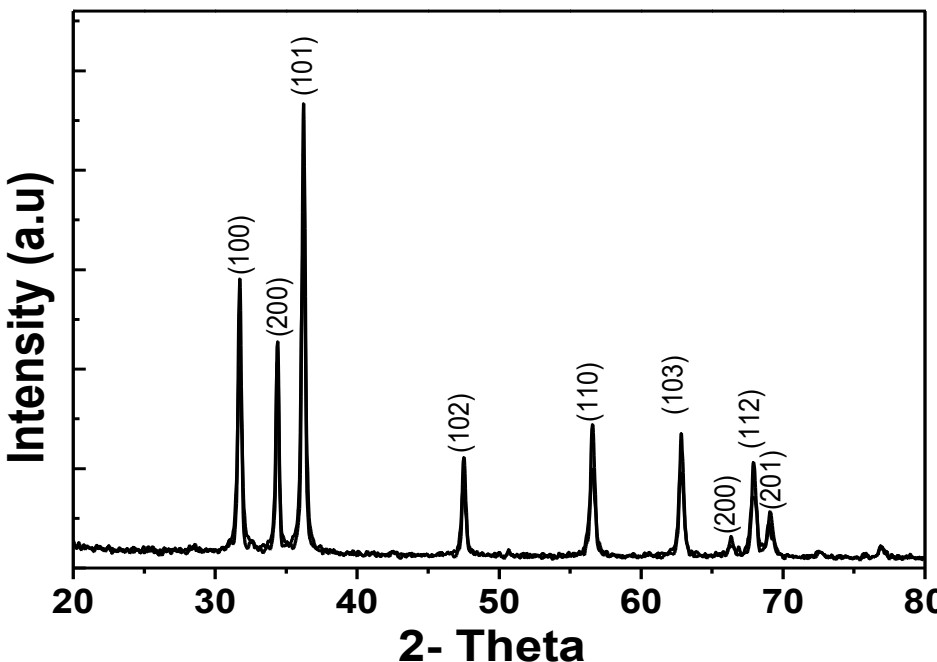

**Figure 6.** Results XRD patterns of ZnO.

### 3.2.2. EDS Analysis

The surface chemical composition of ZnO samples was analyzed by the EDS method. The EDS results show that the surface composition of ZnO is pure, accounting for about 99.99% of Zn and O and only a small amount of carbon impurities (Figure 7 and Table 1). The presence of C can be explained by absorption from the air or incomplete decomposition of organic matter.

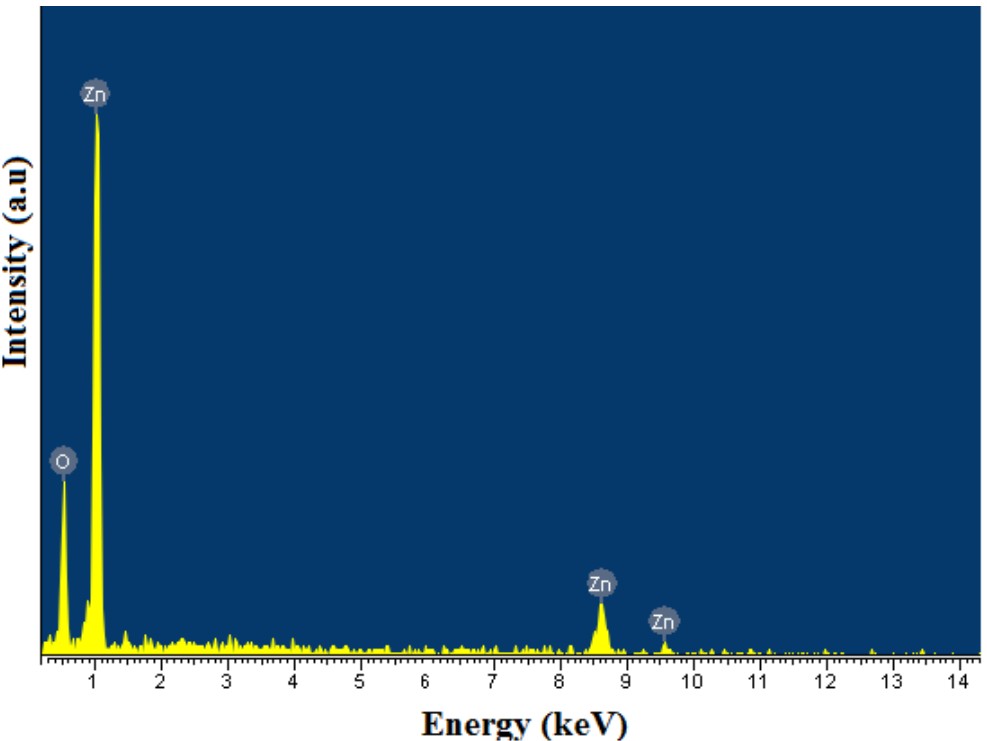

**Figure 7.** EDS pattern of ZnO nanorods.

**Table 1.** Elemental analysis of the synthesized ZnO nanorods.

| Element | Theory | Result |
|---------|--------|--------|
| Zn | 80.34 | 80.32 |
| O | 19.66 | 19.67 |
| Total | 100.00 | 99.99 |

### 3.2.3. Particle-Size Distribution by Laser Diffraction

Particle-size distribution was analyzed by laser diffraction methods (Figure 8). The results indicated that a majority of ZnO nanorods were sized around 0.814 μm, condition synthesis at a temperature of 180 °C. The ZnO particle size measurement range is also small, indicating that the particles synthesized under this temperature condition have a relatively stable size.

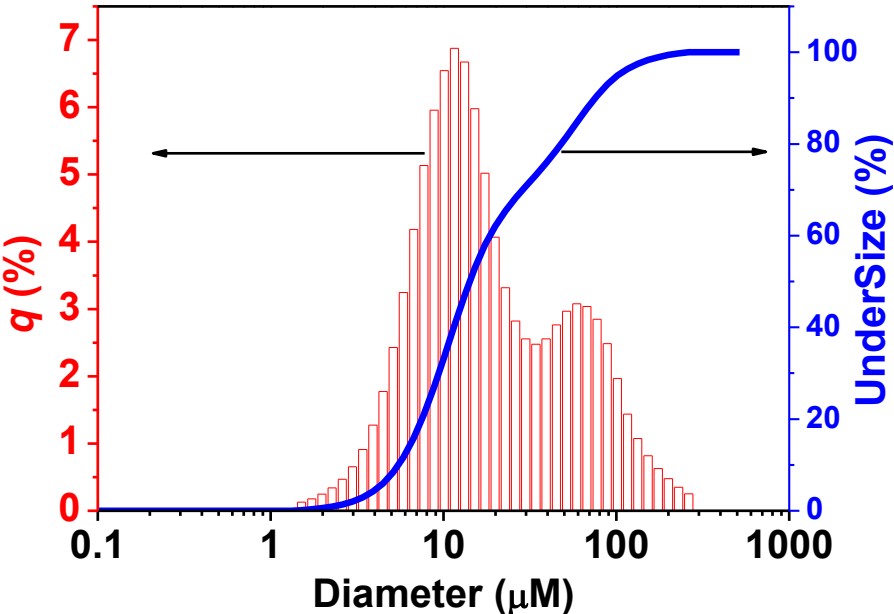

**Figure 8.** Results size distribution of ZnO nanorods by laser diffraction analyzer.

*3.3. Electrochemical Testing*

3.3.1. Cyclic Voltammetry (CV) Curves for Zinc Electrodes

The cyclic voltammetry curves of pure ZnO powder and ZnO nanorods are shown in Figure 9. In Figure 9, pairs of redox peaks and cathode peaks appear. The anodic peak is related to the oxidation reaction of the electrode material during discharge, while the cathode peaks are involved in the opposite process. This redox reaction process of the electrode can be expressed as follows:

$$\text{Charge process: } Zn(OH)_4{}^{2-} + 2e = Zn + 4OH^-$$

$$\text{Discharge process: } Zn + 4OH^- = Zn(OH)_4{}^{2-} + 2e$$

In Figure 9, it is shown that the anodic peak current of the ZnO nanorods is higher, and the anodic peak area is larger than that of the pure ZnO powder, which indicates that the ZnO nanorods have a higher electrochemical activity than that of the pure ZnO powder. The main reason may be that the ZnO nanorods increase the surface area, thereby improving the electron transfer capacity, the electrochemical activity of the redox reaction, and the efficient use of the material. As seen in Table 2 and Figure 9, the anode and cathode peaks are shown. The anodic peaks of ZnO powder and the cathode peaks of ZnO powder are located at $-1.181$ and $1.563$ V, respectively. The anodic peaks of ZnO nanorods and the cathode peaks of ZnO nanorods can be observed at $-1.186$ and $1.545$ V, respectively. ZnO nanorods showed a higher negative anodic peak potential and a higher positive cathodic peak potential than ZnO powder. This indicates that the ZnO nanorods electrode has a higher electrochemical redox reaction process.

The potential interval ($\Delta E_{a,c}$) between the anodic peak potential and the cathode peak potential ($\Delta E_{a,c} = |E_a - E_c|$) is considered to be a measure of the reversibility of the electrode (redox pair) reaction. The smaller the potential interval, the better the reversibility will be. In Table 2, $\Delta E_{a,c}$ for ZnO powder and ZnO nanorod are $0.372$ V, and $0.340$ V, respectively, which indicates that the reversibility of the ZnO nanorod is better than that of ZnO powder.

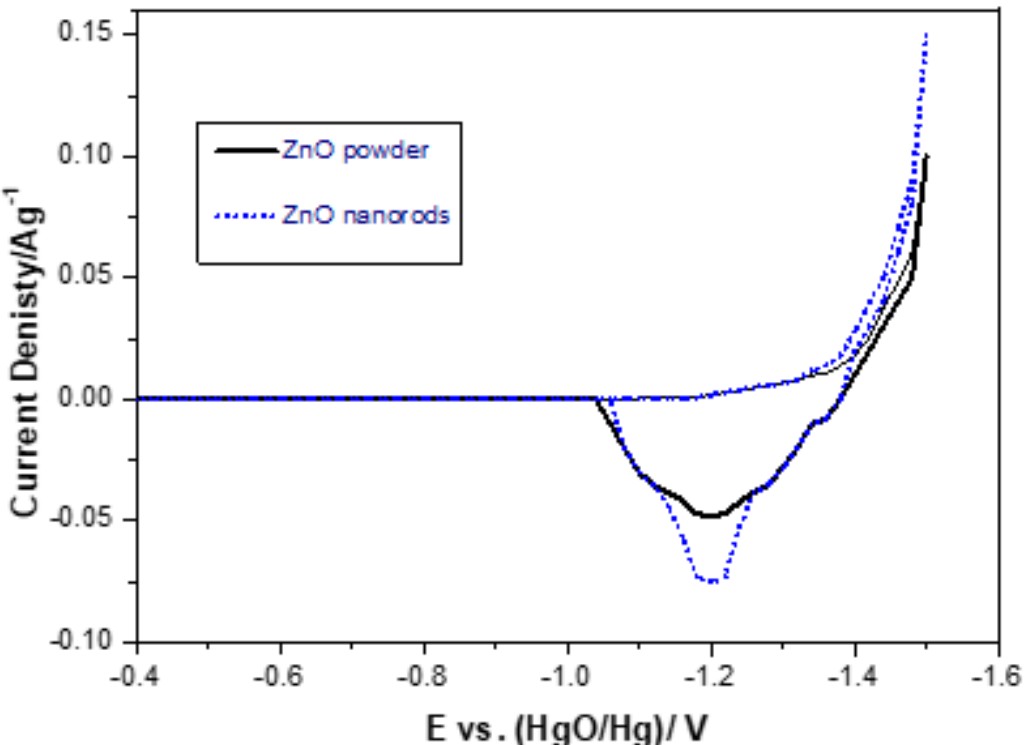

**Figure 9.** Cyclic voltammograms.

**Table 2.** The data of the CV curves for zinc electrodes with ZnO powder and ZnO nanorods.

| Samples | $E_a(V)$ | $E_c(V)$ | $\triangle E_{a,c}(V)$ |
|---------|----------|----------|------------------------|
| ZnO powder | −1.181 | −1.563 | 0.372 |
| ZnO nanorods | −1.186 | −1.546 | 0.340 |

3.3.2. Tafel Plots

Evaluation and investigation of corrosive properties of zinc electrodes by Tafel test method. The Tafel polarization curves of ZnO powder and ZnO nanorods electrodes are shown in Figure 10, and the corresponding corrosion parameters are listed in Table 3. The inhibition efficiency obtained according to Equation (1) is as follows:

$$\text{J.E.\%} = ((J^0_{corr} - J_{corr})/J^0_{corr}) \times 100\% \tag{1}$$

where $J^0_{corr}$ and $J_{corr}$ are the corrosion current of the ZnO powder electrode and the ZnO nanorods electrode, respectively. In Table 3, the ZnO nanorods electrode shows a more positive corrosion potential ($E_{corr}$) and a smaller corrosion current density ($J_{corr}$) than that of the ZnO powder electrode. This indicates that the anti-corrosion performance of ZnO nanorods is better than that of pure ZnO powder. Furthermore, with regard to the surface area of ZnO nanorods, it has a higher surface area, which can increase hydrogen evolution overpotential, thereby reducing the electrode polarization, leading to a positive displacement $E_{corr}$ and a smaller $J_{corr}$.

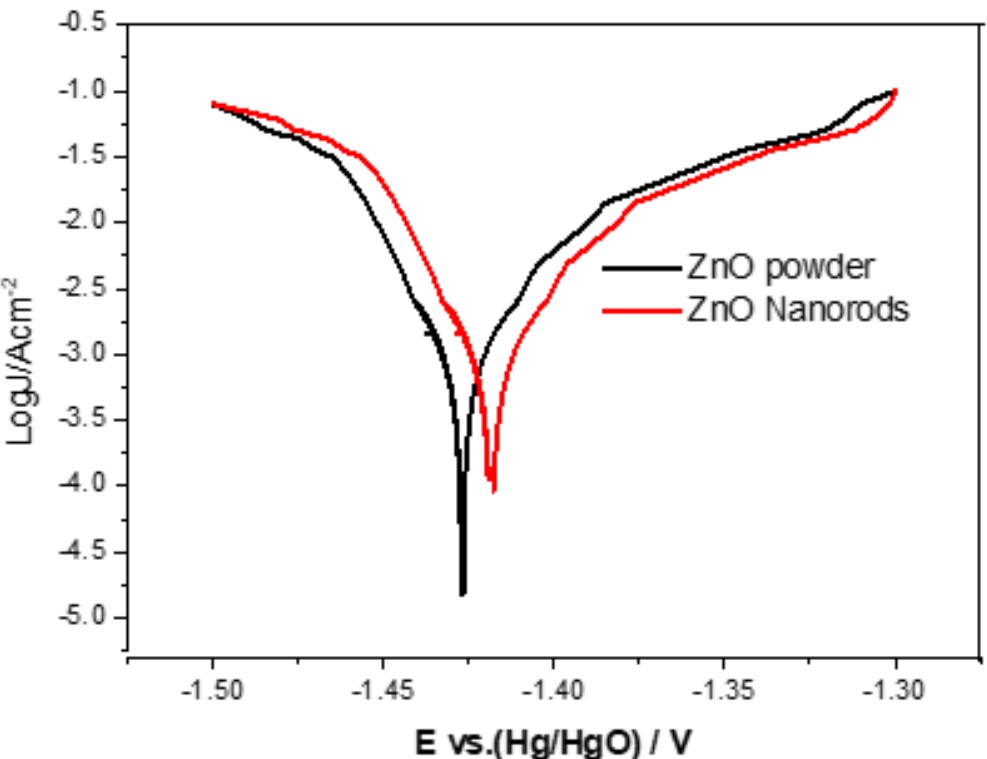

**Figure 10.** Tafel curves for zinc electrodes with ZnO powder and ZnO nanorods.

**Table 3.** The data for the Tafel curves for zinc electrodes with ZnO powder and ZnO nanorods.

| Samples | $E_{Corr}(V)$ | $J_{corr}(mA.cm^{-2})$ |
|---|---|---|
| ZnO powder | −1.4265 | $4.95 \times 10^{-2}$ |
| ZnO nanorods | −1.4180 | $3.86 \times 10^{-2}$ |

### 3.3.3. Properties of Charge/Discharge

The charge/discharge performance of the ZnO nanorod electrodes was investigated by assembling them into a prismatic battery with a capacity of 2.0 Ah. The cell was charged at 0.1 °C and discharged at 0.5 °C for 30 cycles, voltage battery between 1.80 V and 1.350 V. For comparison; the battery was also fabricated using ZnO powder with an average particle size of about three micrometers. The charge/discharge properties of the ZnO powder and ZnO nanorod electrodes are shown in Figure 11. The results show that the cell fabricated with ZnO nanorods shows better open-circuit potential and discharge. At the first cycle, ZnO nanorods are higher than ZnO micro powder and retain their stability until the 30th cycle. The cell fabricated with ZnO micro powder shows a less stable cycle life and capacity of less than 80% after the 30th cycle.

In comparison, the cell fabricated with ZnO nanorods showed higher lifecycle stability. It yielded more than 95% capacity in the 25th cycle (Figure 11). The reason for the enhanced stability of ZnO nanorods is that the solubility of 1D ZnO nanorods is reduced, and the structural stability is enhanced. This reduced solubility and enhanced structural integrity led to the prevention of conformational change and the formation of zinc needles even after the 25th cycle, thereby preventing cell short-circuit and providing lifecycle stability and improving capacity retention.

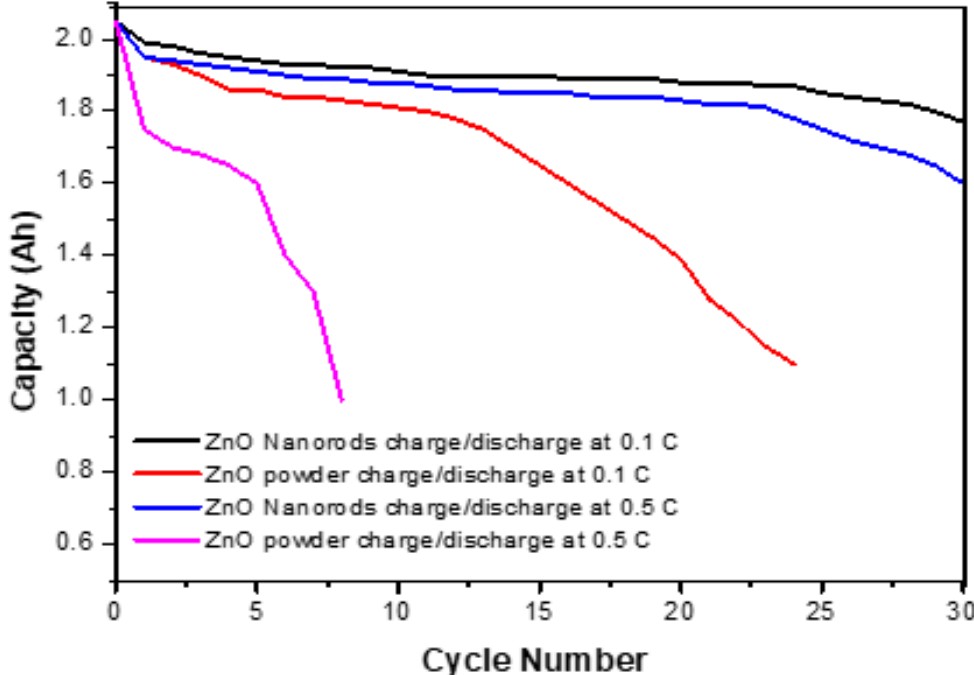

**Figure 11.** Cycle life comparison of ZnO nanorods vs. ZnO micro powder at charge/discharge 0.1 C and 0.5 C.

Figure 11 shows a comparison of the cycle life stability of ZnO micro powders and ZnO nanorods. The ZnO powder cell is stable up to the 10th cycle, but then it degrades and finally fails at the 25th cycle. This is the result of the gradual conformational change of the electrodes and dendritic needle-like formation in powdered ZnO cells. In comparison, it was shown that in the first 25th cycle, the cell fabricated with ZnO nanorods showed higher discharge capacity than the ZnO powder battery and maintained its capacity. This leads to a significant improvement in battery cycle life and breakdowns.

### 3.3.4. Electrochemical Impedance Spectroscopy (EIS)

The better cyclic performance of ZnO nanorods compared with that of ZnO powder electrodes is credited to its lower resistance. Electrochemical impedance spectroscopy (EIS) measurements were collected in the frequency region between 0.01 Hz and 100 kHz with a 5 mV amplitude voltage (Autolab Potentiostat 30) at an open circuit voltage of 1.55 V. Figure 12 shows the Nyquist plots of ZnO powder and ZnO nanorods anode and equivalent circuit. According to the literature [16], $R_e$ represents the total resistance of the electrolyte, electrode and separator. $R_f$ and $CPE_1$ are related to the diffusion resistance of $Zn^{2+}$ ions through the solid electrolyte interface (SEI) layer and the corresponding constant phase element (CPE). $R_{ct}$ and $CPE_2$ correspond to the charge transfer resistance and the corresponding CPE. Zw is related to the solid-state diffusion of $Zn^{2+}$ ions in the active materials corresponding to the sloping line at the low frequency (so-called Zw is the Warburg impedance). The fitting results of $R_e$, $R_f$ and $R_{ct}$ are exhibited in Table 4, indicating that the $R_f$ and $R_{ct}$ values of ZnO nanorods anode are smaller than that of ZnO powder. It can be seen that the resistance of the ZnO nanorods anode is considerably lower than that of the ZnO powder cathode; this is the probable reason for the improved kinetic behaviors during the charge/discharge process. This implies that the presence of ZnO nanorods improves the surface.

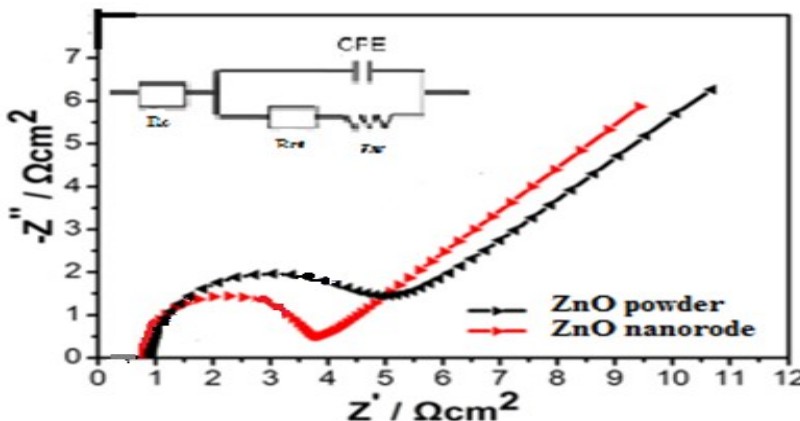

**Figure 12.** Electrochemical impedance spectroscopy plots of the Zn/KOH/Ag$_2$O battery, ZnO nanorods electrodes, with 10 mHz to 100 kHz, amplitude 5 mV and at an open circuit voltage of 1.55 V.

**Table 4.** Impedance parameters calculated from equivalent circuits.

| Samples | $R_e/\Omega$ | $R_f/\Omega$ | $R_{ct}/\Omega$ |
|---|---|---|---|
| ZnO powder | 10.55 | 120.86 | 265.60 |
| ZnO nanorods | 9.45 | 60.81 | 98.78 |

Overall, this work provides the possibility of zinc alkaline batteries using the ZnO nanomaterial. A distinct slope change comes up depending on the diffusion of the ions. Figure 12 shows that the Nyquist spectrum has a semicircle at high to medium frequency mainly involved in a complex reaction at the cathode/electrolyte region. The ramp braking in the lower frequency region is attributed to the Warburg impedance, which is related to the diffusion of Zn$^{2+}$ ions in the ZnO porous electrode. For the slope of ZnO, the strength of ions spread is enhanced; for ZnO nanorods, the strength of ions spread is inhibited. So a conclusion can be deduced that ZnO nanorods materials inhibit the dissolution of ZnO and offer enough contact area to the even deposition of Zn, which corresponds to the analysis results of the cycle life. This implies that the presence of the ZnO nano electrode materials would prove its great potential as a promising for zinc-silver batteries.

## 4. Conclusions

ZnO nanorods were fabricated using the hydrothermal method and characterized with several techniques, including XRD, EDS, BET and laser diffraction. Obtained ZnO nanoparticles under optimal synthesis conditions had a diameter of 50 nm and a length of 100 to 200 nm. ZnO showed a wurtzite structure with a specific surface area of 17.05 m$^2$/g, and its purity reached 99%. As-synthesized ZnO nanoparticles were used in zinc-silver batteries. The prepared materials are used as anode materials for zinc-silver batteries to reduce the shape change of the electrodes. The ZnO nanorods can efficiently improve both the electrochemical cycle stability and the discharge capacity of the anode electrode at a high charge/discharge rate. Compared with the increased BET, the effect of ZnO nanorods on improving the anti-corrosion performance plays a more dominating role in improving the electrochemical performances of ZnO powder anode at the low charge/discharge rate. The charge/discharge results indicate that the nanorods retain their shape and performance throughout the 25th cycle. This improved mechanical behavior leads to improved cycle life, energy density, and higher capacity of zinc-sliver batteries.

**Author Contributions:** V.T.N. and H.T.N. conceived and designed the experiments. H.T.N. and V.T.N. analyzed the data. N.H.T. contributed to the drafting and revision of the manuscript. V.T.N. supervised the work and finalized the manuscript. All authors have read and agreed to the published version of the manuscript.

**Funding:** This work is jointly supported by the National Nature Science Foundation of Vietnam (No.104.06-2017.62).

**Institutional Review Board Statement:** Not applicable.

**Informed Consent Statement:** Not applicable.

**Data Availability Statement:** Not applicable.

**Conflicts of Interest:** The authors declare no conflict of interest.

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
