# Peer review of "Synthesis of ZnO Nanorods and Its Application in Zinc-Silver Secondary Batteries"

_2673-3293, doi:10.3390/electrochem4010008_

Round 1
Reviewer 1 Report (Previous Reviewer 4)
I think the manuscript is now suitable for publication. I can not find any monita now.
Author Response
Thank you very much for your good advice and hard work. We have revised the whole manuscript carefully according to your nice comment. Please find below the Response to Reviewers’ Comments and the revised version.
Reviewer 2 Report (New Reviewer)
see attachment

Author Response
Thank you very much for your good advice and hard work. We have revised the whole manuscript carefully according to your nice comment. Please find below the Response to Reviewers’ Comments and the revised version.

Reviewer 3 Report (New Reviewer)
1. Authors, please provide Figure 3 elemental mapping (EDX) images to support Table 1 details.
2. Please improve the all-figure quality
3. Please check the grammar and correct/revise the mechanism equations to eliminate errors.
4. Suggesting authors to provide after-cycling EIS data to support the concept of the work. If possible, compare before and after cycling EIS data for better understanding.
5. Manuscript required a thorough check to improve writing quality.
6. Please add more related recent journals related to
“Electrochemical Properties of Silver-Zinc Secondary Batteries” in the introduction section.

Author Response
Thank you very much for your good advice and hard work. We have revised the whole manuscript carefully according to your nice comment. Please find below the Response to Reviewers’ Comments and the revised version.

Round 2
Reviewer 2 Report (New Reviewer)
Accept in present form.
This manuscript is a resubmission of an earlier submission. The following is a list of the peer review reports and author responses from that submission.
Round 1
Reviewer 1 Report
The current manuscript demonstrates the electrochemical performance of ZnO for Silver - Zinc Secondary Battery application. The manuscript suffers from lack of novelty. A large quantity of articles have already been published on this topic. Further, results are not properly explained. Rather than a full length manuscript, it looks like a report. The synthesis process is not new. Overall, there is nothing new in this article.
Reviewer 2 Report
- The introduction needs to improve significantly. Several statements and specific information aren’t referenced properly. Here are some of examples.
- “Zinc-silver batteries have high specific energy (up to 300 Wh/kg) and volumetric energy density (up to 750 Wh/dm), low self-discharge rate (~5% per month) and stable voltage during the discharge.”
- “The limitations of zinc electrodes are poor rechargeability, where performancelimiting phenomena are attributed to dendrite growth, shape change, passivation, and hydrogen evolution (as in Figure 2).”
- “ZnO powder is dissolved in alkaline electrolyte and causes the dendritic phenomenon.” Need explanation and reference.
- Experiment section: There is lack of experimental information in this section. For examples, the source of material with purity must be provided. Experiment procedure and characterization conditions must also be described in detail. I recommend the author follow the guideline from the reference. (DOI: doi.org/10.1016/j.jpowsour.2020.227824)
- In Table 1. What are the presented value? That is atomic ration between Zn and O obtained from EDS?
- Section 3.1.1 and Section 3.2.3. The author described the ZnO particles: “the zinc oxide sample has a nanostructure, length of 100 to 200 nm, the diameter of about 50 nm.” What is nanostructure of this materials? The SEM image shows material are unlikely a nanorods. Please explain. In section 3.2.3, Please explain how the majority of particles were determined.
Also, the particle size distribution from laser diffraction analyzer how largely difference from SEM image. Explanation is needed.
- Section 3.3.1 The author need to confirm the value provided in this statement.”The anodic peaks of ZnO powder and the cathode peaks of ZnO powder are located at -1.181, 1.563 V, respectively. The anodic peaks of ZnO nanorods and the cathode peaks of ZnO nanorods can be observed at -1.186, 1.545 V, respectively.” How hes the Ec at Table 2 has positive value?
- The Caption in Figure 9 is unacceptable.
- What specific information has been compared in this statement? At the first cycle, ZnO nanorods are higher than that of ZnO micropowder and retain its stability until the 30th cycle.
- There are several assumptions in explaining the improved performance of cathode with ZnO nanorod. For examples, “The reason for the enhanced stability of ZnO nanorods is that the solubility of 1D ZnO nanorods is reduced and the structural stability is enhanced.” There is no supporting evidence for this statement unless the author can provide the concentration of Zn2+ in electrolyte after cycling with ZnO electrode and post mortem analysis of electrode after cycling. Another examples,” The discharge/charge results indicate that the nanorods retain their shape and performance throughout the 25th cycle”.
Reviewer 3 Report
Following points should be addressed before it is reconsidered for publication.
(1) There are a lot of grammatical errors throughout the manuscript. Please revise it through a professional editing service.
(2) The previous works regarding ZnO anode in aqueous battery systems need to be cited and the originality of this work should be more thoroughly suggested in the Introduction.
(3) "ZnO nanorods limit this disadvantage, thus limiting the dendrite..." in the Introduction needs more explanation. How do nanorods limit the dendrite formation ?
(4) "The results indicate a significant improvement in the shape change..." in the Introduction: what do authors mean by "improvement in the shape change" ?
(5) Figure 3 "length of 100 to 200 nm" : not clear in the figure. Please suggest another picture to show more clearly the length the nanorods.
(6) Please specify the "software" used for the BET plot (Fig. 4)
(7) Anodic peaks in Figure 9 are not clear. Please suggest the magnified plot.
(8) Please give the detailed information of the ZnO powder (particle size, surface area, etc.)
(9) The origin of the anti-corrosion performance of ZnO nanorods need to be more thoroughly discussed with the related references.
(10) Figure 11 (the discharge/charge properties) is missing. Figures 11 and 12 must be Figures 12 and 13, respectively.
(11) Equivalent circuit suggested in Nyquist plot does not match with the text.
Reviewer 4 Report
Review of MS „Synthesis and Electrochemical Performance of ZnO Nanorods…“ (Nguyen et al.) for electrochem
The topic is timely and very interesting for readers of electrochem.
The spectroscopic and electrochemical methods used to characterize ZnO nanorods are appropriate and produce meaningful results. In detail: SEM and laser diffraction show that the synthesized ZnO nanorods are nanoparticles, BET measurements lead to the determination of the active surface area.
The comparison of ZnO nanorods with ZnO powder using the CV method leads to a small improvement of the electrochemical properties such as the reduction of the potential difference between anodic and cathodic current peaks. However, it is difficult to analyze the anodic peak because the evolution of hydrogen occurs in this potential range.
Please correct Ec (V) in Table 2: The values are negative! And: Correct "Talfel" in "Tafel" (page 7).
Page 8: "ZnO nanorods are higher" ... in what? Capacity?
Regarding EIS spectroscopy: The resistances in Figure 12 are illegible. Add to this: In 3.3.4 two constant phase elements CPE1 and CPE2 are found, but they are not further discussed. And Rf is identified as diffusion through the SEI, but I don't think that EIS can distinguish between Warburg diffusion and Rf. Therefore, the interpretation of the EIS results (page 10) seems to me to be an overinterpretation ("the strength of ion propagation is inhibited in ZnO nanorods"), because the slopes in the diffusion region of the EIS spectra of ZnO nanorods and ZnO powder are more or less the same....
In summary, the MS is suitable for publication in electrochem. However, the comments described must be taken into account.
Reviewer
